# The Impact of Shared Leadership on Team Creativity in Innovation Teams—A Chain Mediating Effect Model

**Muyun Sun [1],\*, Kaiyuan He [2],\* and Ting Wen [3]**

[1]    School of Marxism, Nanjing University of Aeronautics and Astronautics, Nanjing 211106, China
[2]    Business School, Hohai University, Nanjing 210024, China
[3]    College of Business Administration, Nanjing University of Finance & Economics, Nanjing 210023, China
\*    Correspondence: muyun2027@163.com (M.S.); he_ky@hhu.edu.cn (K.H.)

**Abstract:** As an important outcome of team innovation, team creativity has become an important issue in academia and industry. Meanwhile, the horizontal leadership model has been preliminarily proven to be effective in improving the output of innovation performance. Multiple chain mediating effects of team psychological safety climate, cognitive motivation and social motivation on shared leadership and team creativity in innovative teams were proposed on the basis of social network theory and group dynamics theory. In this study, 178 innovation teams and 2011 innovation team members were given questionnaires, and the obtained data were empirically analyzed. The results show that shared leadership has a significant positive effect on team creativity in innovative teams; team psychological safety climates, cognitive motivation and social motivation play a partial mediating role between shared leadership and team creativity, and play a chain mediating role together. At the team level, the study verifies the positive effect of shared leadership on team creativity and reveals the complex team process.

**Keywords:** shared leadership; team creativity; team psychological safety climate; cognitive motivation; social motivation

## 1. Introduction

"Innovation is the primary force guiding development". As innovation creativities become increasingly complex, innovative teams have shown better performance over individuals and have gradually become a major form of innovation activities in organizations. Team creativity, as an important yield of innovative teams, has already become a major issue in academia and industry. Team leadership has already been proven to be an important factor influencing team output, while the traditional top-down vertical leadership model has revealed various disadvantages in innovative teamwork. Compared with the vertical leadership model, which is characterized by relatively stable leading roles, decentralized and transferable leadership may give full play to the advantages of individuals with key information and knowledge in specific circumstances, and as such, a new leadership model is called shared leadership [1]. An increasing number of scholars have paid attention to the synergy of "multi-source leadership" [2]. Indeed, shared leadership has been noticed and adopted by an increasing number of companies; for instance, Huawei's team management model and Xiaomi's matrix management model, both of which adopt the horizontal collective leadership model.

Although there have been a large number of studies on leadership models and team creativity in academia, most only focus on the impact of vertical leadership models on team creativity, while few have noticed the horizontal leadership model; further explorations are still needed. Existing research is still controversial on whether shared leadership is the decisive factor of team performance, but it is still believed that the impact of team processes such as team coordination and knowledge sharing on team output can be studied [3]. Meanwhile, shared leadership, which originates in the West, has yet to reveal whether

it can enhance the creativity of innovative teams and how it can enhance team creativity in the Chinese context [4]. On this basis, this paper analyzes the influence of the special leadership network formed by shared leadership in innovation teams on team creativity according to the social network theory and group dynamics theory. It is expected to further explore the specific influence path of shared leadership on team output, as well as the practical possibility in the Chinese management context.

## 2. Research Hypotheses

The innovation team that adopts the shared leadership mode forms a special network of horizontal influence among members within the team. This special network changes the objective relationship and network form among members, thus exerting direct or indirect influences on the internal atmosphere of the team, the motivation of the overall information processing of the team and the creativity of the team. Specific hypotheses are as follows.

A hypothetical model for studying the effect of shared leadership on team creativity in innovative teams is shown in Figure 1.

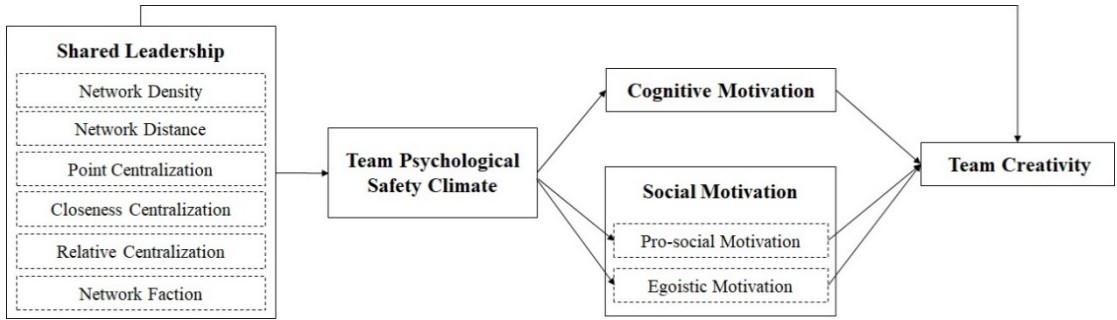

**Figure 1.** Hypothetical Model.

(1) Effect of shared leadership on team creativity in innovative teams

Proposed by Kurt Lewin, an American social psychologist, group dynamics is a "field theory" of human behavior [5]. According to group dynamics, the primary internal drivers of a group come from the cohesion, driving force and dissipative force, which coexist in the group and convert into, contend with and digest each other to promote the development of the group, and the formation of hierarchy within a group can have an impact on team performance [6]. In view of innovative teams, knowledge sharing has been proven to be an important foundation for teams to achieve innovation [7]. From the perspective of group dynamics, the relationship between team members, who are the main subjects of knowledge sharing, is a decisive factor in achieving effective knowledge dissemination [8], and team members shall provide others with accurate knowledge in innovative work, or seek useful knowledge from others. Research has pointed out that when team members clearly know "who has the knowledge and expertise in need", the "distribution of knowledge and expertise" and other issues, the knowledge network of the group or team is effective. Team members can accurately seek from others or provide others with the knowledge in need [9], and shared leadership provides teams with an internal network that describes individual awareness of expertise. Previous studies have demonstrated a close relationship between internal network characteristic indexes and team creativity [10]; for instance, network centrality, weak ties, communication and interaction with "different people" are all conducive to creativity [11]. The social network is a common method of network membership analysis, and Perry-Smith et al. described how social networks bring together dispersed knowledge that can lead to creative solutions, and emphasized the significance of the impact of other people on creativity [12]. Henceforth, they integrated the theories of creativity and social network and explored the direct and indirect effect of network distance, network density, network location and external relationship on creativity [13]. Tang Chaoying and Huang Dongling put forward the influence path

of internal team network on R&D creativity by reviewing domestic and foreign research literature from the perspective of internal team network relationships, network structure and node characteristics [14]. Hargadon and Bechky built a model of collective creativity, focusing on the process of how a group of creative individuals becomes a creative collective [15]. It was found that individuals with novel changes may arouse collective creativity by seeking others' help. It is a mode of feedback to offer assistance, which can bring about high-level expectations and more positive assessments, and farsightedness is usually presented through seeking new similarities and establishing new connections [16]. This kind of collective creativity is known as team creativity in an innovation team, which is defined as the new ideas generated by team members in the process of achieving the team goals [17]. Huang and Liu also found that colleagues' assistance and support can not only strengthen the relationship between knowledge heterogeneity and staff creativity, but also enhance the relationship between network relationship and staff creativity [18]. However, the development of shared leadership network factions inevitably leads to small groups within the team, among which the competition and ties would generate dissipative forces and reduce team creativity. In general, innovation teams under shared leadership display a low degree of centrality concentration (the value of centrality is relatively high) in the leadership network and decentralized authority, which can ensure the dominant role of team members at work and allow team members to feel full trust and respect, and further arouse the initiative and enhance the creativity of individual members. Particularly, in the Chinese management context, this is characterized by a "relationship orientation"; the degree of centrality in innovation teams of shared leadership is low because there may be multiple leading centers in a team and the leading power is shared among team members, which can inspire and encourage team members and further enhance team creativity. Therefore, the following hypothesis is proposed:

**Hypothesis 1.** *Shared leadership has a positive effect on team creativity in innovation teams.*

(2)  Team psychological safety climate, cognitive motivation and social motivation

Edmondson et al. interpreted team psychological safety at the organizational level as the collective belief formed in an organization by its members about the ability to take interpersonal risks in the organization without worrying about being threatened [19]. Specifically, when the psychological safety climate in the organization is relatively high, team members are more willing to make suggestions to the organization or seek help from other team members, admit mistakes at work actively, or come up with new ideas actively, etc. It is generally agreed that the above behaviors may have negative effects on the individual image and status in the organization, or hinder vocational development, and can even cause more damages. Baer and Frese further extended the concept of psychological safety by combining its definition with organizational practices and considering psychological safety as interactions that may guide and support individuals in the organization to trust each other [20]. Accordingly, psychological safety at the organizational level tends to express more of a psychological perception, in which members do not worry about punishment or even retaliation for speaking freely. Essentially, the team psychological safety climate can be understood as an internal and invisible common belief and collective perception, and also the usual way of doing things within the team that is commonly accepted by team members. However, team members shall go through the same organizational procedures and management practices to develop such a common belief and collective cognition, and eventually develop unanimous values and attitudes that are recognized and agreed upon by all members [21].

Aiming at information processing, cognitive motivation and social motivation are proposed in the motivational information processing theory [22]. Specifically, cognitive motivation is defined as the willingness to make efforts to understand a certain thing, while social motivation means the preference for a particular distribution of outcomes between oneself and the other, which can be divided into pro-social and egoistic motivation. When

a team has a low cognitive motivation, it tends to solve problems with heuristic strategies, hoping to spend less time to produce results, thus being superficial and unable to further explore the things [23]. On the contrary, when a team has a high cognitive motivation, it tends to spend lots of time and effort in evaluating the problems and gathering information in order to dig up information and knowledge and to develop a better plan [24]. Therefore, cognitive motivation directly affects the depth of information processing in the process of information processing [25]. The pro-social motivation includes cooperative tendency and altruistic tendency. Driven by pro-social motivation, individuals wish to maximize their own and others' interests and they not only focus on their own interests, but also hold an active concern for the interests of their opponents, which is easy to form a win-win situation. Egoistic motivation includes competitive tendencies and selfish tendencies. Egoistic-motivated individuals wish to maximize their own interests, without considering others' interests. When innovative teams display a high degree of shared leadership, a high network density and low centralization, team members would develop strong emotional ties and a high perception of fairness. According to the research conclusions of De Dreu and other scholars, it is easy to arouse the cognitive motivation of team members, and team members can fully exchange knowledge to spend more time and efforts in discussing problems rigorously and meticulously, thus inspiring team creativity [26]. Based on the group dynamics theory, when team members develop a closer relationship, the network of relationships formed inside a team can enhance the perception of trust among team members, and more resources will be passed among team members [27]. Moreover, increased interaction can lead to the establishment of the same team cognition among team members, which not only leads to greater identification with team goals, but also increases the sense of identity and dependence on other members. Furthermore, this strong altruistic sentiment can arouse the pro-social motivation of team members who are more willing to pay for other team members and the whole team.

On the whole, shared leadership allows members of innovation teams to develop more connections and closer relationships with each other, to have stronger emotional ties, and to have more trust and emotional attachment to each other. All of these factors may arouse the cognitive and pro-social motivation of team members and reduce egoistic motivation [28]. Therefore, the following hypotheses are proposed:

**Hypothesis 2.** *Team psychological safety climate has a positive impact on cognitive motivation.*

**Hypothesis 3.** *Team psychological safety climate has a positive impact on pro-social motivation.*

**Hypothesis 4.** *Team psychological safety climate has a negative impact on egoistic motivation.*

(3)　Cognitive motivation, social motivation and team creativity

Motivation is deemed as a major factor driving creativity. Although motivational information processing theory was proposed in the context of western culture, it has been supported by many empirical studies in China. It has been proposed that when a team has low cognitive motivation, it will have a negative impact on knowledge acquisition, and team members will not spend extra time and efforts in enriching the team knowledge [29]. In contrast, when the tasks to be completed are appealing, or individuals need to undertake more responsibilities for decisions, cognitive motivation would be inspired [30]. Therefore, challenges posed by creative work in innovation teams inspires the job competency of intellectual workers, and they will dig deeper into knowledge and information to improve team creativity [31].

Pro-social and egoistic motivations are much like the collectivism and individualism concept in Chinese culture, and it has been proven that individuals with a high sense of collectivism have higher and more conspicuous creative performance than individuals with individualistic ideas [32]. Baston proposed that employees with higher pro-social motivation are more willing to pay for others and organizations, care more about the

interests of others and organizations and are more likely to improve their performance due to the willingness to give [33]. Grant also gave some explanations to pro-social motivation in his study, arguing that individuals with high pro-social motivation would take actions on the premise of helping others or organizations and making contributions to others, and they are driven by these ideas [34]. Grant also further discovered that specific contexts can also inspire the pro-social motivation; for instance, good organizational climate, harmonious interpersonal relationships, awareness of fairness, etc. would inspire team members to help others and improve the organizational performance [35]. Egoistic motivation is also interpreted in the Chinese management context as having an individualistic tendency. Team members tend to maximize their own interests, and sometimes ignore or even devalue the results of others. In competitions, people with egoistic motivation will take their own success as crucial and even tend to undermine the interests of others. Pro-social and egoistic motivation may co-exist and interact to affect the whole team, and form the motivational tendency of the entire team.

In the research on cognitive motivation and social motivation, members of innovative teams can be regarded as information processors, and both cognitive and social motivations would affect the choices of members in the process of information processing from different perspectives. According to the motivational information processing theory, cognitive motivation affects the depth of information processing, while social motivation affects the breadth of information processing. A team can only maximize its creativity when both motivations are high. Combined with the actual work situation, cognitive motivation means the willingness to spend more time and efforts in digging up information, and indeed, individuals with higher cognitive motivation would spend more time on work, concentrate on work and complete the innovative work more efficiently. Pro-social motivation reflects the willingness to help others, and in practice, it is presented as team members' consideration for the output of the team and team members in the creative work. Moreover, they are more willing to contribute their knowledge, skills and other resources. At this moment, individuals with higher pro-social motivation may help the team produce higher creativity. In contrast, individuals with high egoistic motivation think more about themselves regardless of others, and they fail to contribute to the team, which may reduce team creativity [36].

Therefore, the following hypotheses are proposed:

**Hypothesis 5.** *Cognitive motivation has a positive impact on team creativity.*

**Hypothesis 6.** *Pro-social motivation has a positive impact on team creativity.*

**Hypothesis 7.** *Egoistic motivation has a negative impact on team creativity.*

(4)　Chain mediation

The characteristics of shared leadership formed inside the team, such as high network density, low network distance, low centralization of network points, etc. can help to enhance the direct connections and form emotional ties between team members, increase the team psychological safety climate, and allow the team to gain positive emotions. Mutual trust and tolerance among team members help to enhance the overall cognitive and pro-social motivation of the team and reduce the egoistic motivation. When team members have a higher psychological safety climate, they are willing to trust other members instead of worrying that different voices may affect their status or reputation in the team. At this moment, cognitive motivation has been stimulated, and individuals or the team as a whole may concentrate more on the further exploration of information, sufficient sharing of knowledge and acquisition of deeper tacit knowledge, thus improving the creativity of both the team and individuals [19]. It has been noted that higher psychological safety enables team members to focus more on the areas of expertise of others, and consider more about other members and the team. As a result, the pro-social motivation is boosted, while

the egoistic motivation is reduced. The increased tolerance and openness may help with the acquisition of broader knowledge and improve the creativity [37].

It has been proposed in research on social motivation that personal traits have a strong impact on pro-social motivation; for instance, individuals with higher collectivism tendency tend to have a high pro-social motivation, and individuals with low agreeableness have higher egoistic motivation. However, the social motivation of individuals may change due to the impact of external environment. Leaders are the shapers of team climate, and a high degree of shared leadership may develop a fair and just climate inside the team, which can improve the psychological safety climate of team members, develop positive emotional perceptions, increase the pro-social motivation and reduce the egoistic motivation. The increase in pro-social motivation can drive both individuals and the team to share information and knowledge on a larger scale, and further enhance the creativity of both individuals and the team. Therefore, the following hypothesis is proposed:

**Hypothesis 8.** *Team psychological safety climate, cognitive motivation and social motivation have a chain intermediation effect on shared leadership and team creativity of innovative teams.*

### 3. Research Design

#### 3.1. Sample Selection and Data Sources

This research aims to explore the relationship between shared leadership and team creativity in innovative teams. The social network analysis and research with teams as the research object were needed to obtain the complete information of a team, and each team member was involved in the survey. Therefore, convenient sampling, rather than random sampling, was adopted in most cases because objects of random sampling may not come from the same team, which may not develop a social network with boundaries [38]. Because the objects of the survey were innovation teams, and sampling units were knowledge-intensive organizations, this research eventually decided to choose the innovation teams from colleagues, universities, scientific research institutions and enterprises as the research objects. One of the main research variables of this research is shared leadership. In China's innovation teams, there are few teams formally proposing to adopt the shared leadership model, but according to Carson's definition of shared leadership, when team members perceive that other team members play an informal leading role in the team, shared leadership is formed in the team. Therefore, before the formal survey, in addition to the innovation teams that explicitly propose the use of shared leadership, those that do not practice shared leadership were also studied. Through the interviews with team leaders and members, the internal working style and process of the team were understood for judging if shared leadership had been practiced by the team; if so, such teams would also be surveyed.

Due to the large sample size required for team-level research and the great difficulty of collecting research variables including network characteristic variables and requiring the cooperation between team leaders and all team members, this research selected regions and organizations with certain social connections to conduct the survey in order to guarantee the efficiency and quality of the survey. The formal large-scale survey started in October 2021 and ended in December 2021. A total of 200 sets of questionnaires including 2481 copies were successively distributed to innovation teams in 10 universities, 6 scientific research institutions and 7 enterprises in Nanjing, Hangzhou, Shanghai, Suzhou, Qingdao, Chengdu and Hong Kong. The whole survey lasted for nearly two months, recovering 2237 questionnaires from 198 innovation teams, with a 90.17% recovery rate. The questionnaires were distributed on-site and online. To guarantee the quality of questionnaires, most were distributed on-site. Generally, data of teams with missing samples would be screened in order to ensure the validity of recovered questionnaires, and the usual standard is 80% of recovery rate. Over 20% of missing questionnaires would affect the network situation of the whole team; missing questionnaires of key figures would also affect the network situation of the team [39]. Therefore, this research adopted this standard and eliminated the

questionnaires of teams whose recovery rate was lower than 80%. Eventually, we acquired 2011 valid questionnaires from 178 innovation teams, with an efficiency rate of 90.00%. Finally, all valid questionnaires gathered were sorted out according to the basic situation of sample teams.

### 3.2. Variable Measurement

Measurement of shared leadership. Previous studies that evaluated shared leadership with the social network analysis method mostly chose network density and network centralization as the indexes for measurement [40], which were combined to reflect the distribution of leading powers among team members. Network density is generally interpreted as an index indicating the closeness of relationships among team members; network centralization is used to measure the degree of dependence of team members on other members. In this research, it was insisted that shared leadership not only includes the closeness and dependency between team members, but also shows the distribution of leadership in teams, the mutual horizontal influence of team members and the internal leadership structure. Therefore, the use of network density and network centralization may not be able to fully show the characteristics of shared leadership. Consequently, this research proposed to choose network distance, closeness centralization, relative centralization and network faction to fully express the shared leadership of teams, so as to essentially reflect shared leadership. Network distance is the average shortest distance between n nodes in a network, which reflects the network cohesion to a certain extent. Network point centralization reflects the central tendency of nodes in a network. Network closeness centralization reflects the difference of nodes in a network. The higher the closeness centralization is, the greater the difference will be. Network intermediate centralization reflects the concentration of overall network resources (whether information depends on a node for transmission); the higher the intermediate centralization, the more concentrated the network resource. Faction analysis is an index reflecting the network aggregation coefficient, which can be used to judge whether there are small groups in the network. In adopting a social network approach for measuring shared leadership, this research chose the classic Mayo question for measuring shared leadership: has a non-team formal leader ever had a leadership influence on you in your innovation team? If so, to what extent do you rely on his leadership? (1 point, "not at all"; 7 points, "to a great extent") Specific network indexes were calculated using Ucinet 6.0 software.

Team creativity measurement. This research applied questionnaire rating to measure creativity. This research adopted Farh and Lee's development scale to measure the team creativity. This scale is modified on the basis of the creativity questionnaire developed by Oldham and Cummings. The scale contains three items, and questions for measurement include "our team's output is creative", etc. The Cronbach's $\alpha$ of the scale in this research was 0.833.

Team psychological safety climate. Team psychological safety climate is a relatively new area of research, and the scale developed by Edmondson is the one of the most frequently used scales. This research also chose this scale, and the questions include "it's safe to take risks in the team", etc. The Cronbach's $\alpha$ of the scale in this research was 0.754.

Cognitive motivation. Scales measuring the cognitive motivation are relatively unified. This research chose the cognitive motivation scale used for the creativity study by Bechtoldt, De Dreu and Choi. It was also used by Yang Shupeng, a Chinese scholar. The questions include "I always think hard, trying to come up with more and better ideas", etc. The Cronbach's $\alpha$ of the scale in this research was 0.780.

Social motivation. This research chose the measurement scale used by Beersma and De Dreu to test the social motivation, and the questions include "I attempt to make myself more profitable in team work.", etc. The Cronbach's $\alpha$ of the scale in this research was 0.810.

Control variables. Based on previous studies, team size and the team stage were adopted as the control variables in this research.

## 4. Data Analysis and Hypothesis Testing

### 4.1. Data Aggregation

Because team psychological safety climate, cognitive motivation, social motivation and team creativity are obtained by aggregating individual evaluations, it was necessary to do the aggregation testing. The results are shown in Table 1, where the Rwg (reliability within the group) is over 0.7, the ICC (1) (inter-group correlation) is over 0.2 and ICC (2) is over 0.6, which meets the judgment conditions for performing team-level data summation [41].

**Table 1.** Results of intergroup variation tests for each variable.

| Variables | Average Value of Rwg | ICC (1) | ICC (2) |
|---|---|---|---|
| Team Psychological Safety Climate | 0.903 | 0.417 | 0.690 |
| Cognitive Motivation | 0.897 | 0.303 | 0.787 |
| Social Motivation | 0.927 | 0.369 | 0.726 |
| Team Creativity | 0.880 | 0.416 | 0.738 |

### 4.2. Discriminant Validity and Common Method Bias Test

(1) Discriminant validity. In this research, Mplus 7.0 software was used for confirmatory factor analysis to test the discriminant validity among the constructs. Confirmatory factor analysis was conducted for the benchmark model (four-factor) containing team psychological safety climate, cognitive motivation, social motivation and creativity, and alternative factor models (one-factor, two-factor and three-factor) were constructed for comparison with the hypothesized factor model; the results are shown in Table 2. According to the results, the indexes of the four-factor model all met the requirements of a qualified model, which far exceeded other alternative factor models, suggesting the good discriminant validity of the research variables.

**Table 2.** Results of confirmatory factor analysis.

| Model | $\chi^2$ | $df$ | $\chi^2/df$ | CFI | TLI | RMSEA |
|---|---|---|---|---|---|---|
| Single Factor Mod (TPS + CM + SM + TC) | 6023.201 | 1491 | 4.040 | 0.677 | 0.747 | 0.051 |
| Two Factor model (TPS + CM + SM, TC) | 5779.073 | 1477 | 3.913 | 0.792 | 0.798 | 0.044 |
| Three Factor model (TPS + CM, SM, TC) | 3136.190 | 1463 | 2.144 | 0.893 | 0.913 | 0.041 |
| Four Factor model (TPS, CM, SM, TC) | 2603.991 | 1456 | 1.788 | 0.915 | 0.918 | 0.027 |

Note: TPS for team psychological safety climate, CM for cognitive motivation, SM for social motivation, TC for team creativity, and the same below.

(2) Common method bias. Because the data of all variables came from the same survey respondents, one-way factor analysis was used for the common method variance test of data. The results showed that the rotation generated a total of four factors, which explained a total of 76.889% of the total variance, exceeding the critical limit of 70%, and the variation explained by the first factor was 25.455%, lower than the requirement of 50%, indicating that there was no serious common method variance problem in this research.

### 4.3. Descriptive Statistics and Correlation Coefficients between Variables

Descriptive statistics analysis of variables and correlation coefficients were conducted using SPSS 22.0 software, and the results are shown in Table 3. The results showed clearly that most of the variables are correlated with each other, which preliminarily indicates the relationship as described in the hypotheses and provides a necessary prerequisite for further analysis.

**Table 3.** Descriptive statistical analysis and correlation coefficients of variables.

| Variables | 1 | 2 | 3 | 4 | 5 | 6 | 7 | 8 | 9 | 10 | 11 | 12 |
|---|---|---|---|---|---|---|---|---|---|---|---|---|
| 1. Team Size | 1 | | | | | | | | | | | |
| 2. Team Stage | 0.014 | 1 | | | | | | | | | | |
| 3. Network Density | 0.017 | 0.101 | 1 | | | | | | | | | |
| 4. Network Distance | 0.126 | 0.108 | 0.111 ** | 1 | | | | | | | | |
| 5. Point Centralization | 0.129 | 0.256 | 0.250 ** | 0.217 * | 1 | | | | | | | |
| 6. Closeness Centralization | 0.216 | 0.122 | 0.268 ** | 0.042 | 0.223 ** | 1 | | | | | | |
| 7. Relative Centralization | 0.047 | 0.227 | 0.155 * | 0.198 * | 0.115 ** | 0.103 ** | 1 | | | | | |
| 8. Network Faction | 0.233 * | 0.301 | 0.211 ** | 0.115 | 0.194 | 0.251 ** | 0.122 * | 1 | | | | |
| 9. Team Psychological Safety Climate | 0.031 | 0.077 | 0.086 * | −0.046 | 0.034 ** | 0.075 * | 0.225 * | −0.098 * | 0.916 | | | |
| 10. Cognitive Motivation | 0.101 | 0.054 | 0.242 ** | −0.032 | 0.081 * | 0.018 | 0.134 * | −0.172 | 0.191 ** | 0.881 | | |
| 11. Social Motivation | 0.055 | 0.207 | 0.104 * | 0.122 * | 0.034 * | 0.094 * | 0.093 * | 0.119 * | 0.155 ** | 0.050 | 0.890 | |
| 12. Team Creativity | 0.041 | 0.091 | 0.272 ** | 0.253 ** | 0.119 ** | 0.140 * | 0.012 | −0.040 | 0.191 ** | 0.176 * | 0.167 ** | 0.803 |
| Mean Value | 3.309 | 3.738 | 0.359 | 1.524 | 0.472 | 0.515 | 0.214 | 3.3 | 5.095 | 5.717 | 4.605 | 5.036 |
| Standard Deviation | 0.745 | 0.663 | 0.186 | 0.294 | 0.300 | 0.212 | 0.207 | 1.302 | 0.225 | 0.981 | 1.110 | 1.170 |

Note: ** $p < 0.01$, * $p < 0.05$.

### 4.4. Hypothesis Testing

(1) Direct effect test. Multiple logistic regression is a statistical technique [42]. In this study, Mplus 7.0 software was used to perform multiple logistic regression to test each hypothesis path. With team creativity as the dependent variable and shared leadership as the independent variable, the results showed that shared leadership was positively correlated with team creativity ($\beta = 0.783$, SE = 0.036, $p < 0.001$), and hypothesis 1 was supported. With cognitive motivation and social motivation (pro-social motivation and egoistic motivation) as dependent variables and team psychological safety climate as the independent variable, the results showed that team psychological safety climate was positively correlated with cognitive motivation ($\beta = 0.863$, SE = 0.047, $p < 0.001$), positively correlated with pro-social motivation ($\beta = 0.742$, SE = 0.028, $p < 0.001$), and negatively correlated with egoistic motivation ($\beta = -0.663$, SE = 0.059, $p < 0.001$), thus hypotheses 2, 3 and 4 were supported. With team creativity as the dependent variable and cognitive and social motivation (pro-social and egoistic motivation) as independent variables, the results showed that team creativity was positively correlated with cognitive motivation ($\beta = 0.773$, SE = 0.041, $p < 0.001$), positively correlated with pro-social motivation ($\beta = 0.719$, SE = 0.020, $p < 0.001$), and negatively correlated with egoistic motivation ($\beta = -0.658$, SE = 0.067 $p < 0.001$), thus hypotheses 5, 6 and 7 were supported. The results of model path analysis are shown in Figure 2.

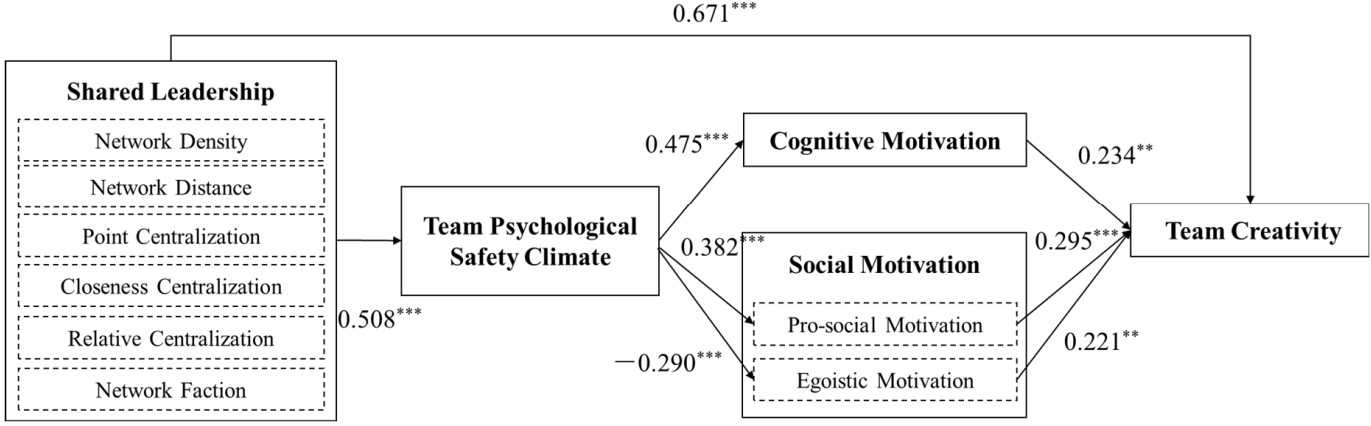

**Figure 2.** The Results of Model Path Analysis. Note: *** $p < 0.001$, ** $p < 0.01$.

According to the results of the direct effect test, in the innovation team, shared leadership can directly and effectively improve the team creativity. Meanwhile, shared leadership objectively makes the relationship between team members closer, creates a good team atmosphere and effectively improves the overall team psychological safety climate. The team psychological safety climate can effectively stimulate the cognitive motivation and pro-social motivation of the whole team and reduce the egoistic motivation of the whole team. The improvement of the team's overall cognitive motivation and pro-social motivation can promote the team's information processing and directly improve the creativity of the innovation team. The reduced egoistic motivation also contributes to the information processing and improves the team's creativity.

(2) Chain mediating effect test. At present, studies on mediation effects have developed from simple mediation to multiple mediation, in which Bootstrap has become a common research method [43]. Multiple chained mediating effects in the hypotheses were tested using Mplus 7.0 based on Bootstrap analysis method. Sampling with replacement was conducted 5000 times for valid samples, and eucalyptus approximating sampling distribution of the total mediating effect and the specific mediating path effect was obtained. The confidence intervals of the mediating effect with 95% of confidence at the 2.5 percentile point (LLCI) and the 97.5 percentile point (ULCI) were constructed; when the interval does not contain 1, it indicated a significant mediating effect. According to Table 4, the effect value of shared leadership → team psychological safety climate→ team creativity is 0.229, with a 95% confidence interval {0.213,0.395}, which does not contain 0, and the indirect effect is significant; the effect value of shared leadership → cognitive motivation→ team creativity is 0.261, with a 95% confidence interval {0.191,0.418}, which does not contain 0, and the indirect effect is significant; the effect value of shared leadership→ pro-social motivation→ team creativity was 0.306, with a 95% confidence interval {0.233,0.494}, which does not contain 0, and the indirect effect was significant; the effect of shared leadership→ egoistic motivation→ team creativity was 0.205, with a 95% confidence interval {0.182. 0.377}, which does not contain 0, and the indirect effect is significant; the effect value of shared leadership→ team psychological safety climate→ cognitive motivation→ team creativity is 0.054, with a 95% confidence interval {0.017,096}, which does not contain 0, and the chain mediating effect is significant; the effect value of shared leadership→ team psychological safety climate→ pro-social motivation→ team creativity is 0.085, with a 95% confidence interval {0.058,0.126}, which does not contain 0, and the chain mediating effect is significant; the effect value of shared leadership → team psychological safety climate → egoistic motivation → team creativity is 0.031, with a 95% confidence interval {0.012,075}, which does not contain 0, and the chain mediating effect is significant, and hypothesis 8 was supported.

**Table 4.** Results of chain mediating effect test.

| Full Mediation Model | Normalized Path Coefficient | | | | | Point Estimation | Confidence Interval | |
|---|---|---|---|---|---|---|---|---|
| | TPS | CM | PM | SM | TC | (Non-Standardized) | Lower Limit | Upper Limit |
| SL | 0.508 *** | | | | 0.671 *** | | | |
| TPS | | 0.475 *** | 0.382 *** | −0.290 ** | | | | |
| CM | | | | | 0.234 ** | | | |
| PM | | | | | 0.295 *** | | | |
| SM | | | | | 0.221 ** | | | |
| SL→TPS→TC | | | | | | 0.229 | 0.213 | 0.395 |
| SL→CM→TC | | | | | | 0.261 | 0.191 | 0.418 |
| SL→PM→TC | | | | | | 0.306 | 0.233 | 0.494 |
| SL→SM→TC | | | | | | 0.205 | 0.182 | 0.377 |
| SL→TPS→CM→TC | | | | | | 0.054 | 0.017 | 0.096 |
| SL→TPS→PM→TC | | | | | | 0.085 | 0.058 | 0.126 |
| SL→TPS→SM→TC | | | | | | 0.031 | 0.012 | 0.075 |
| Total Mediating Effect | | | | | | 0.671 | 0.337 | 0.709 |

Note: Bootstrap = 5000, SL is for shared leadership. *** $p < 0.001$, ** $p < 0.01$.

According to the results of the mediating effect test, this study shows and confirms the mediating influence paths of "shared leader-team psychological safety climate—team creativity", "shared leader-cognitive motivation-team creativity", and "shared leader-social motivation-team creativity". The results show that shared leadership can create a good team climate, thus improving team creativity; in addition, shared leadership can also improve the motivation of team information processing, thus improving team creativity. More importantly, this study constructed and tested the complete chain mediation of "shared leader-team psychological safety climate—team cognitive motivation—team creativity", "shared leader-team psychological safety climate—team pro-social motivation—team creativity", and "shared leader-team psychological safety atmosphere—team egoistic motivation—team creativity". Chain mediation further explains the complex mechanism process that shared leaders promote for the motivation of information processing by improving the team psychological security climate, thus improving the team creativity.

## 5. Research Conclusions and Discussion

### 5.1. Research Conclusions and Contribution

This research observed the direct and indirect effect of shared leadership on team creativity, and drew the following conclusions: shared leadership enhances the overall psychological safety climate, cognitive motivation and social motivation of innovation teams. That is, it not only improves the pro-social motivation, but also reduces the egoistic motivation and enhances team creativity, in addition to team psychological safety climate and cognitive motivation. Furthermore, social motivation plays a mediating role in the relationship between shared leadership and team creativity. The mechanism of shared leadership's effect on team creativity in innovation teams is revealed, and how shared leadership plays a positive role in innovation teams is explained; in addition, the influence mechanism of shared leadership on team creativity is more completely elaborated. Especially in the Chinese management context, when the demand for psychological perception of knowledge workers is met, their motivation for information processing will be better stimulated, which is more conducive to the completion of innovation-oriented tasks and the output of higher creativity.

The theoretical contribution of this research lies in that ① It enriches research on the impact of shared leadership on team creativity. Although several studies have examined the relationship between shared leadership and team creativity, few consider the networks constructed by shared leadership within teams or examine the impact of leadership networks on team creativity from the perspective of the social network. This research verifies the positive impact of network density of shared leadership on team creativity based on the social network theory, and extends the network measure of shared leadership, proving that network point centralization, intermediate centralization and closeness centralization can improve team creativity. The exploration provides considerations and assumptions about the characteristics of group network resources theoretically and expands the methods of network research nodes methodologically for the horizontal leadership model. ② It proposes and tests the chain mediating effect of team psychological safety climate, cognitive motivation and social motivation on shared leadership and team creativity. The special leadership network developed by shared leadership within an innovation team creates a stronger team psychological safety climate in the team, which enhances the overall team cognitive motivation, and team members are more willing to dig up knowledge, share knowledge and reduce egoistic considerations in a more secure context, thus creating higher team creativity. This is in response to some scholars' suggestion that shared leadership in some special situations, such as innovation-oriented situations, is more conducive to exerting advantages [1]. Similarly, more detailed analysis of the underlying action mechanism of shared leadership on team creativity has been launched in which further response enriches relevant studies proposed by Charles et al. in different cultural settings and provides more empirical evidence [2].

### 5.2. Practical Implications

The practical implication of this research is that organizations and innovation teams shall admit the value of shared leadership in the improvement of team creativity, and practice shared leadership actively in management practice. In China, a good interpersonal relationship plays a vital role in team operations. If there is a good interpersonal relationship network in the team, it will facilitate the team operation and improve the team output. Shared leadership essentially helps to develop a good network in the team, which not only creates more communication channels to facilitate communication and information exchange between leaders and members as well as between team members, but also builds a good interpersonal network. Therefore, we shall manage to maximize the advantages of shared leadership in management practice, enhance cultural construction, develop a good atmosphere in the team, promote the exchanges between team members, and guarantee the improvement of team creativity.

For leaders and managers, they shall make team members feel that they are pleased to see the participation in team management. It is a way to bring out the talents of team members by increasing participation, to stimulate their enthusiasm by empowering them with leading power, and to strengthen their individual effectiveness by increasing their autonomy. Shared leadership also has disadvantages, so it is necessary to avoid its negative impact in practical management. Therefore, team leaders with rich management experience are needed to make plans for team development stages and specific team tasks in practical management situations, and pay attention to the effect of vertical leadership on the team while using the shared leadership model. Team leaders shall let team members with leading talents take a leading role in the team and avoid personal interest cliques inside the team so that experts of each field can not only fully exert their expertise without affecting the improvement of team creativity, but also achieve team outputs.

### 5.3. Research Limitations and Future Prospects

Shared leadership is a relatively new research topic in the field of leadership, with a relatively small research base but complex issues. Although this research enriches the relevant theoretical research, it still has shortcomings that need to be improved due to the limitation of objective conditions: ① Owing to the difficulty in collecting team questionnaires and resource limits, this research chose cross-sectional data, and all data were collected at the same time point. In particular, chain mediation hypothesis was proposed in this research, and there may be doubts on the casual relationship between research variables. To avoid this problem, this research carried out a logical derivation on the basis of the S-O-R model and creativity atmosphere–motivation model, hoping to avoid the possibility of causal inversion. In future research, longitudinal studies, experimental studies and case studies can be taken into account to further explore the mechanism of influence between shared leadership and team creativity. ② Samples are not evenly distributed. Due to the resource limits, the innovation teams of this research were chosen from colleges and universities, research institutes and enterprises (mainly private enterprises in the IT industry and communication industry), and there is a lack of samples from state-owned enterprises and other organizations. As a whole, stratified sampling was not fully conducted for samples of this research to ensure the comprehensiveness of innovation teams. In future surveys, organizations of innovation teams shall be diversified.

**Author Contributions:** Conceptualization, M.S.; methodology, M.S. and K.H.; software, M.S. and K.H.; validation, T.W.; formal analysis, M.S.; investigation, M.S. and T.W.; resources, M.S.; data curation, M.S.; writing—original draft preparation, M.S.; writing—review and editing, K.H. and T.W.; visualization, M.S.; supervision, K.H.; project administration, M.S.; funding acquisition, M.S. All authors have read and agreed to the published version of the manuscript.

**Funding:** This work was funded by the Basic Scientific Research Funds of Universities (1023-XCA22007).

**Institutional Review Board Statement:** The study was conducted in accordance with the Declaration of Helsinki, and approved by the Institutional Review Board (or Ethics Committee) of INSTITUTE OF HUMAN RESOURCES, MINISTRY OF WATER RESOURCES (30 June 2021).

**Informed Consent Statement:** Informed consent was obtained from all subjects involved in the study.

**Data Availability Statement:** The data presented in this study are available on request from the corresponding author (Muyun Sun, muyun2027@163.com). The data are not publicly available due to some raw data about the research subjects need to be approved.

**Conflicts of Interest:** The authors declare no conflict of interest.

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
