# Peer review of "The Impact of Shared Leadership on Team Creativity in Innovation Teams—A Chain Mediating Effect Model"

_sustainability, doi:10.3390/su15021212_

Round 1
Reviewer 1 Report
The article is written in a clear and professional manner. It includes an adequate analysis of the literature sources and discusses the entire research procedure in a detailed and professional manner, along with the presentation of the results and the conclusions drawn from them. In the reviewer's opinion, with the removal of minor comments, the work can be submitted for publication.
The main limitations of the article are as fallows:
1. In the literature section of the paper, the authors refer to two theories: thw social network theory and group dynamics theory. Although these are well-known concepts in the literature, the authors should provide their definitions and main assumptions, or at least the sources in which they are described in detail.
2. On page two of the article, the authors use the phrase "Research has pointed out that ..." - at this point, the exact source to the results of the research discussed should be cited because otherwise the reader has no way to verify the cited results and research conclusions.
3. The last remark concerns hypothesis number 1. In order to verify such a research hypothesis, it is necessary to clearly and precisely define the term "team creativity." Otherwise, the process of its verification will be exposed to a high degree of subjectivity in the evaluation (anyone will be able to define the term in question in any way found in the literature). This is all the more important because the concept of creativity appears in the scientific literature in many different contexts and approaches.
Author Response
The Impact of Shared Leadership on Team Creativity in Innovation Teams——A Chain Mediating Effect Model
(Manuscript No.: 2082748)
We thank the anonymous reviewers for their valuable comments, which certainly help to improve quality of the manuscript. The manuscript has been revised accordingly, and the reviewer comments are addressed below.
Response to Reviewer 1 Comments
The article is written in a clear and professional manner. It includes an adequate analysis of the literature sources and discusses the entire research procedure in a detailed and professional manner, along with the presentation of the results and the conclusions drawn from them. In the reviewer's opinion, with the removal of minor comments, the work can be submitted for publication.
The main limitations of the article are as fallows:
Point 1: In the literature section of the paper, the authors refer to two theories: thw social network theory and group dynamics theory. Although these are well-known concepts in the literature, the authors should provide their definitions and main assumptions, or at least the sources in which they are described in detail.
Response 1: Thank you for your comments. We have supplemented the relevant definitions and sources of group dynamics and social network theory, as well as the relevant literature (page 3, line 67-72; page3, line 86-87).
Point 2: On page two of the article, the authors use the phrase "Research has pointed out that ..." - at this point, the exact source to the results of the research discussed should be cited because otherwise the reader has no way to verify the cited results and research conclusions.
Response 2: Thank you for your comments. The research article has been cited (page 3, line 81),
Point 3: The last remark concerns hypothesis number 1. In order to verify such a research hypothesis, it is necessary to clearly and precisely define the term "team creativity." Otherwise, the process of its verification will be exposed to a high degree of subjectivity in the evaluation (anyone will be able to define the term in question in any way found in the literature). This is all the more important because the concept of creativity appears in the scientific literature in many different contexts and approaches.
Response 3: Thank you for your comments. We have We have added the definition of team creativity in H1 (page 3, line 101-103). Team creativity is defined as follows: the new ideas generated by team members in the process of achieving the team goals.

Reviewer 2 Report
The paper is interesting, even if not original at all. The analysis is linked at a specific country with particular features compared to those of many other countries. Yet it can be a good reference point for similar studies to be conducted in other countries. A more adequate introduction and literature review could justify the research. Introduction presents properly the aim of the study, yet the research questions to be addressed are not clearly exposed and, above all, justified by the literature. As a matter of fact, the authors must include accurate and recent references to support the hypotheses and the study. So, strongly I suggest to consider a more recent and innovative papers on the topic and important in the international context. Research design and methodology could be appropriate, yet different analyses have been conducted which enrich the empirical analysis (so again, the authors must consider further literature, like Doi 10.1002/csr.1873): I recommend the authors to better specify the goodness of the specific quantitative method to support the conceptual model. And moreover, why is the used methodology better than other important ones? And besides, are the authors sure that the sample is representative of the population? Especially interesting is the analyses conducted, but I can say also the results could be more appropriate and clear; moreover, discussion section is relevant and conclusions must resume properly the topic address and the implications for several players. So, really what does the paper add to previous researches? The quality of communication is good and clear enough.
Author Response
The Impact of Shared Leadership on Team Creativity in Innovation Teams——A Chain Mediating Effect Model
(Manuscript No.: 2082748)
We thank the anonymous reviewers for their valuable comments, which certainly help to improve quality of the manuscript. The manuscript has been revised accordingly, and the reviewer comments are addressed below.
Response to Reviewer 2 Comments
Point 1: The paper is interesting, even if not original at all. The analysis is linked at a specific country with particular features compared to those of many other countries. Yet it can be a good reference point for similar studies to be conducted in other countries.
A more adequate introduction and literature review could justify the research. Introduction presents properly the aim of the study, yet the research questions to be addressed are not clearly exposed and, above all, justified by the literature. As a matter of fact, the authors must include accurate and recent references to support the hypotheses and the study. So, strongly I suggest to consider a more recent and innovative papers on the topic and important in the international context.
Response 1: Thank you for your insightful comments. We have added new literature in the introduction and literature review, and the specific supplementary literature is as follows:
[1] Ning Xu, Hamed Ghahremani, G. James Lemoine, Paul E. Tesluk. Emergence of shared leadership networks in teams: An adaptive process perspective[J]. The Leadership Quarterly, 2022,12(33):101588
[2] Charles George, Cristina B.Gibson, Jennifer Barbour. Shared leadership across cultures: Do traditionalism and virtuality matter?[J] Journal of International Management, 2022,1(28): 100905
[3] Zhao Jingjing. An Empirical study on innovative services for library reading promotion based on dynamic reading——Taking Wenzhou Library reading drama society as an example[J]. Library Work and Study, 2022(10):111-115
[4] Wellman, N., Applegate, J. M., Harlow, J., et al. Beyond the pyramid: Alternative formal hierarchical structures and team performance[J]. Academy of Management Journal, 2019, 63(4) .
[5]Gao, D., Akbaritabar, A. Using agent-based modeling in routine dynamics research: A quantitative and content analysis of literature[J]. Review of Managerial Science, 2021, 10: 1007/s11846-021-00446-z.
Point 2: Research design and methodology could be appropriate, yet different analyses have been conducted which enrich the empirical analysis (so again, the authors must consider further literature, like Doi 10.1002/csr.1873): I recommend the authors to better specify the goodness of the specific quantitative method to support the conceptual model. And moreover, why is the used methodology better than other important ones?
Response 2: Thank you for your comments. We have added the literature you recommended in 4.4 (page 11, line 405).
Additional explanations are given in the methods used (page 11, line 407-407, page 12, line 439-441).
Point 3: And besides, are the authors sure that the sample is representative of the population? Especially interesting is the analyses conducted, but I can say also the results could be more appropriate and clear; moreover, discussion section is relevant and conclusions must resume properly the topic address and the implications for several players. So, really what does the paper add to previous researches? The quality of communication is good and clear enough.
Response 3: The limitations of the sample are explained in 5.3. In the sampling process, teams from universities and scientific research institutes are taken as the main research objects. In future studies, the samples can be further enriched to make the samples more representative (page 15, line 564-570).
In 5.1, a dialogue with the previous literature is added(page 13-14, line 519-524).

Reviewer 3 Report
The title needs to be changed - rhetorical questions do not look good in a research paper
Abstract gives some info on the results and partially methods - but it is not enough. Please provide some information on the backgrounf and motivation and present results with some important figures or clear outcomes.
Inroduction is short, however it clearly states the background stressing the fact that the problem is planned to be studied through the Chinese context. A lot of background information is given further in the RH section so this introduction design is ok here.
RH section takes up half of the body of the paper and is really detailed, presenting 8 different RHs. It serves as a literature review or related work section so probably such volume is justified. All the RHs are extensively grounded, though just in the form of a suggestion - I feel there is a way to unite some of the RHs and the interrelations into a paradigm or a system rather than those separate 8 RHs - they are all interrelated and deviding them this way somehow prevents from forming a holistic picture of those interrelations - the problem is solved by figure 1 though - however while reading all those pages you feel like you really needed just this figure 1 to get the idea. So if there is no better arrangement for this -it is ok, if you keep the related work within this section probably there is no other way that the one chosen by the authors. Then if you keep the RHs this way - then probably - figure 1 has to go in the beginning of the section - after some brief introduction of all the aspects from Fig. 1, stating that there are such parameters to be considered - show figure 1 - and then start deviding it into all those RHs - start with general - then go to the details - this way it becomes scientific style.
One more aspect needs addressing - a good part of the literature is 10+ years old - I understand there is some fundamental research on the matter, however the background of the research subject in the paper will benefit if authors refer to some latest works.
Research design section is devided in to 2 subsections, is well-structured and is presenting the main stages of the experiment (survey), data processing and other aspects. The section is easy to follow and presents all the main poitns of th design.
Section 4 is devidied in to 4 subsections, each providing the relevant part of the D analysis and H testing. The section is well-structured, tables and figures and informative and clear.
However the following subsection 5.2 on practical implication is vague and I suggest that sectiion 4 is extended with some clear results of the survey in the form of bullet-points or other format clearly stating the outcomes of the survey without using percentage or other calculation results - please state in a practical way - what the results of the survye discover - where are the gaps, or whar are the main interdependencies or interrelations are - referring the shared leadership terms and implications - not figures.
Section 5 is rather descriptive - however being relatively new topic that is acceptable. This section with a lot of reflection (text) looks odd in comparison with section 4 - where results are given in the form of calculations. The topic of the paper is leadership - so after applying the math to your analysis - provide the insights that are connetcted with your math - but talk about the poblem. Section 4 and 5 now seem like two extremities - math and "story" - find a bridge please. We need a "story"- but based on math and related to the solution of the stated problem.
Author Response
The Impact of Shared Leadership on Team Creativity in Innovation Teams——A Chain Mediating Effect Model
(Manuscript No.: 2082748)
We thank the anonymous reviewers for their valuable comments, which certainly help to improve quality of the manuscript. The manuscript has been revised accordingly, and the reviewer comments are addressed below.
Response to Reviewer 3 Comments
Point 1: The title needs to be changed - rhetorical questions do not look good in a research paper.
Response 1: Thank you for your comments. We have changed the title to “The Impact of Shared Leadership on Team Creativity in Innovation Teams——A Chain Mediating Effect Model”.
Point 2: Abstract gives some info on the results and partially methods - but it is not enough. Please provide some information on the backgrounf and motivation and present results with some important figures or clear outcomes.
Response 2: We have added background information and some explanatory results to the abstract.
Point 3: RH section takes up half of the body of the paper and is really detailed, presenting 8 different RHs. It serves as a literature review or related work section so probably such volume is justified. All the RHs are extensively grounded, though just in the form of a suggestion - I feel there is a way to unite some of the RHs and the interrelations into a paradigm or a system rather than those separate 8 RHs - they are all int1errelated and deviding them this way somehow prevents from forming a holistic picture of those interrelations - the problem is solved by figure 1 though - however while reading all those pages you feel like you really needed just this figure 1 to get the idea. So if there is no better arrangement for this -it is ok, if you keep the related work within this section probably there is no other way that the one chosen by the authors. Then if you keep the RHs this way - then probably - figure 1 has to go in the beginning of the section - after some brief introduction of all the aspects from Fig. 1, stating that there are such parameters to be considered - show figure 1 - and then start deviding it into all those RHs - start with general - then go to the details - this way it becomes scientific style.
Response 3: Thank you for your comments. Adding some general explanation at the beginning of the hypothesis really helps the reader understand why the hypothesis is advancing the way it is. So we added a little supplement at the beginning of the Research Hypotheses section and put the hypotheses model here (page 2, line 55-60).
Point 4: One more aspect needs addressing - a good part of the literature is 10+ years old - I understand there is some fundamental research on the matter, however the background of the research subject in the paper will benefit if authors refer to some latest works.
Response 4: We have added recent research articles to the introduction and research hypotheses. The added articles are as follows:
[1] Ning Xu, Hamed Ghahremani, G. James Lemoine, Paul E. Tesluk. Emergence of shared leadership networks in teams: An adaptive process perspective[J]. The Leadership Quarterly, 2022,12(33):101588
[2] Charles George, Cristina B.Gibson, Jennifer Barbour. Shared leadership across cultures: Do traditionalism and virtuality matter?[J] Journal of International Management, 2022,1(28): 100905
[3] Zhao Jingjing. An Empirical study on innovative services for library reading promotion based on dynamic reading——Taking Wenzhou Library reading drama society as an example[J]. Library Work and Study, 2022(10):111-115
[4] Wellman, N., Applegate, J. M., Harlow, J., et al. Beyond the pyramid: Alternative formal hierarchical structures and team performance[J]. Academy of Management Journal, 2019, 63(4) .
[5]Gao, D., Akbaritabar, A. Using agent-based modeling in routine dynamics research: A quantitative and content analysis of literature[J]. Review of Managerial Science, 2021, 10: 1007/s11846-021-00446-z.
Point 5: However the following subsection 5.2 on practical implication is vague and I suggest that sectiion 4 is extended with some clear results of the survey in the form of bullet-points or other format clearly stating the outcomes of the survey without using percentage or other calculation results - please state in a practical way - what the results of the survye discover - where are the gaps, or whar are the main interdependencies or interrelations are - referring the shared leadership terms and implications - not figures.
Response 5: We have expanded 4.4, especially to further explain and illustrate the results of the path analysis (page 12, line 427-437; page 13-14, line 470-483).
Point 6: Section 5 is rather descriptive - however being relatively new topic that is acceptable. This section with a lot of reflection (text) looks odd in comparison with section 4 - where results are given in the form of calculations. The topic of the paper is leadership - so after applying the math to your analysis - provide the insights that are connetcted with your math - but talk about the poblem. Section 4 and 5 now seem like two extremities - math and "story" - find a bridge please. We need a "story"- but based on math and related to the solution of the stated problem.
Response 6: We analyzed and further explained the calculation results in 5.1, hoping to simply play the role of "story" (page 14, line 493-500).

Round 2
Reviewer 2 Report
Now I am satisfied, but I see an error in the reference number 41, the names and the sur names are inverter, the correct form is: Boccia F. and Sarnacchiaro P.
Author Response
We thank the reviewer for his valuable comments, which certainly help to improve quality of the manuscript. The 41 reference has been revised.